# Understanding public health risk from unsafe dry fish consumption in Bangladesh

Mahdi Al Hasan Rahat[1⊙], Anik Saha[1⊙], Mehedy Hasan Abir[1,2⊙], A. S. M. Nafis Sadekeen[1], Shahneaz Ali Khan[3], Sukanta Chowdury[4]*

1 Faculty of Food Science and Technology, Chattogram Veterinary and Animal Sciences University, Chattogram, Bangladesh, 2 Nutritional Sciences Graduate Program, Margaret Ritchie School of Family and Consumer Sciences, College of Agricultural & Life Sciences, University of Idaho, Moscow, ID, United States of America, 3 Department of Physiology, Biochemistry and Pharmacology, Chattogram Veterinary and Animal Sciences University, Chattogram, Bangladesh, 4 International Centre for Diarrhoeal Disease Research, Bangladesh (icddr,b), Dhaka, Bangladesh

⊙ These authors contributed equally to this work.
* sukanta@icddrb.org

**Data Availability Statement:** All relevant data are within the manuscript and its Supporting Information file.

## Abstract

Dried fish holds a significant place in the Bangladeshi diet particularly for people living in coastal regions. However, there is a growing concern regarding its adverse effects on human health, as it contains high levels of illegal preservatives, heavy metals, and other harmful substances. In this study, we aimed to explore the current knowledge, attitudes, and practices regarding health hazards due to unsafe dried fish consumption among people across the country. We conducted a cross-sectional study among consumers to assess their knowledge, attitudes, and practices about the health risks associated with consuming hazardous dried fish. We interviewed a total of 415 participants, of whom 52.8% were male; the majority were students (55.9%), aged between 18 and 30 years (63.9%), and living in urban areas (81.7%). Most of the participants (60.7%) had less accurate knowledge of the health hazards of unsafe dry fish, 92.8% had more positive attitudes to buying safe dry fish, and 26.8% used unsafe dry fish more frequently. Many respondents were unaware of the presence of harmful substances in dried fish, such as illegal pesticides (66.5%), microplastics (77.6%), and heavy metals (67.4%). A significant number of participants (13%) reported that they had a history of cancer in any of their family members. Many individuals (57.4%) were not familiar with the proper storage and preparation methods of dry fish. The majority of participants (81.4%) strongly prefer packed dried fish. Most of the respondents (67.7%) agreed to participate in awareness programs. Female consumers were more likely to have more accurate knowledge (AOR = 1.53; 95% CI = 1.03–2.29, $p$ = 0.0.37) than males, and participants whose present residence were in rural were more likely to have accurate knowledge (AOR = 2.64; 95% CI = 1.30–5.36, $p$ = 0.007) than those whose present residence were in urban or semi-urban areas. A targeted education campaign focused on improving awareness of the risks associated with eating unsafe dry fish is needed, particularly in coastal areas.

**Funding:** The author(s) received no specific funding for this work.

**Competing interests:** The authors have declared that no competing interests exist.

## Introduction

Dry fish serves as a source of protein and it is a typical food item in many places, especially in areas where fresh fish are poorly available [1]. In Bangladesh, the fish sector contributes to the country's economy, nutrition, and earning money from abroad [2]. More than 20% of locally caught fish are sun-dried and consumed locally [3]. Adequate sanitation and hygiene are not maintained properly during drying process of fish. Bug infestation, the presence of dirt, and pesticide residue are commonly found in dried fish [1]. The fish processors are responsible for drying, packing, and distributing fish to markets and they do not have adequate training in sanitation practices [4]. The dried fish is often kept close to the seaside town in a warehouse. Due to blowfly infestation and longer drying times might result in spoilages and quality losses of dry fish. Bug infestation can cause dried fish to lose 30–40% of their weight [4]. There is also the issue that many fish processors prioritize profit over quality, leading to improper drying practices that make insects more prevalent more quickly [5]. Dry fish processors typically used several insecticides such as dichlorodiphenyltrichloroethane (DDT), aldrin, acephate, diazinon and dimethoate to prevent insects [6, 7]. Pesticides and the low microbiological quality of dried fish are concerning for Bangladeshi consumers since they may cause prolonged sickness and provide long-term harm to human health [8].

Pesticide-contaminated dried fish is unsafe for consumption that poses health risks to humans. Pesticide residues in dried fish may cause cancer, epilepsy, liver and kidney damage, leukemia, decreased fertility, genetic damage, and immune system suppression [9, 10]. The deposition of heavy metals and microplastics from the aquatic environment effect on the quality of dried fish [11]. Upon consumption, heavy metals present in dry fish pose moderate-to-high health concerns to the human body, and dried fish polluted with Pb and Cd pose a cancer risk to consumers [7, 12].

The attitudes and behaviors of food consumers and handlers influence their degree of knowledge of food safety [13]. Consumers' mindset is a key element that affect their behavior and practices with regard to food safety [14, 15]. A positive outlook can help to suggest safety information to ensure that everyone has access to safe food [16]. There have been no previously published KAP studies on consumers in Bangladesh about food safety problems resulting from the consumption of unsafe dry fish. This study focused on exploring the current knowledge and practices regarding health hazards due to unsafe dried fish consumption. This study's findings help to understand consumers' knowledge gaps and identify risky practices.

## Methods

### Study sites, population and study period

A cross-sectional study was carried out throughout different divisions in Bangladesh, with a special focus on the coastal division Chattogram due to its significant quantity of dried fish production and consumption [17], between November and December 2022. A combination of in-person and online interviews were conducted, with the majority of interviews being conducted in person (373 in-person interviews). For the in-person interview, we selected participants purposively who were aged above 18 years. The study team visited a dry fish market every day and invited visitors or buyers or consumers to participate in this study. Participants were interviewed, and data was collected using a structured questionnaire administered by an interviewer, who asked questions and recorded the information. Additionally, a web-based questionnaire was distributed via social media platforms (Facebook, Instagram, LinkedIn, and Gmail) to collect online data, with participants given instructions before filling out the form themselves.

## Data collection

We collected data on socio-demographics, health indicators, consumption habits of dry fish, knowledge, attitudes, and practices on health hazards about unsafe dry fish and associated practices.

a. **Socio-demographic data:** Socio-demographic information included age, gender, religion, division, marital status, education, occupation, monthly family income (BDT), family size, place of present address, place of permanent residence, and frequency of living with the family.

b. **Health indicators:** The history of family health included nine questions to know whether there were previous risks associated with dried fish consumption.

c. **Knowledge, attitudes, and practices:** A total of 39 questions were included to assess the level of knowledge, attitudes, and practices.

After completion of the initial draft of the questionnaire, the questionnaire was validated and adopted by knowledgeable academics and experts. After coordination and consensus from the experts, the questionnaire has been finalized for data collection. The calculated sample size was 385 individuals based on 95% confidence interval, a 5% margin of error, and the assumption that 50% of the respondents are at risk from consuming unhealthy dried fish [18]. The formula is as follows:

$$n_0 = \frac{Z^2 pq}{e^2}$$

Here, $n_0$ = Sample size; Z = Z-score = 1.96 for 95% level of confidence; p = Estimated proportion of the attribute that is present in the population = 50% (or 0.5); q = 1-p = 0.5; e = Margin of error = 5%.

In this study, the we collected data from 415 potential respondents. The reliability coefficient analysis was conducted for collected data. The Cronbach's alpha coefficient of the knowledge, attitudes, and practices were 0.84, 0.70, and 0.70, respectively, and the overall Cronbach's alpha of KAP questions was 0.84, which indicated acceptable internal consistency as described [19].

The knowledge section consisted of 13 questions and each of them had a possible response of "*Yes*", "*No*", and "*Don't know*" (e.g., *Do you think consuming preservative-free dried fish is good for health*?). The correct answer (*Yes*) was coded as 1, while the wrong answer (*No/ Don't know*) was coded as 0. The total score ranged from 0–13, with an overall greater score indicating "*More Accurate*" knowledge. A cut-off level of ≥10 was set for "*More accurate*" knowledge, 6–10 was set for "*Moderate*" knowledge, and ≤5 was set for "*Less accurate*" knowledge.

The attitudes section consisted of 15 questions, and each of them was indicated on a 3-point Likert scale as follows: 0 for "*Disagree*", 1 for "*Undecided*", and 2 for "*Agree*" (e.g., *Consumers should have some knowledge on the hazards of dried fish before buying*). The total score was calculated by summating the raw scores of the six questions ranging from 0 to 30, with an overall greater score indicating "*More positive*" attitudes. A cut-off level of ≥21 was set for "*More positive*", 11–20 was set for "*Moderate*", and ≤10 was set for "*Less positive*" attitudes.

The practices section included a total of 11 questions, and each of them had one of the responses of "*Yes*", "*No*", and "*Sometimes*" (e.g., *Do you buy dried fish from a trusted seller/processor*?). Practice items' total score ranged from 0 to 22, with an overall greater score indicating "*More frequent*" practices. A cut-off level of ≥15 was set for "*More frequent*", 8–14 was set for "*Moderate*", and ≤7 was set for "*Less frequent*" practices.

## Statistical analysis

We summarized participants' demographic characteristics such as age, sex, marital status, education, occupation and residence status by descriptive analyses (frequency, mean, standard deviation, p-value and 95% confidence interval). Chi-square test was performed to examine the differences between categories for each categorical value with respect to their knowledge, attitudes, and practices. Multivariate logistic regression analysis was performed to identify the association between the variables and respondents' knowledge, attitudes, and practices on health hazards due to unsafe dry fish through. We estimated the odds ratio (OR) and adjusted odds ratio (AOR) separately for knowledge, attitudes, and practices. All statistical analyses were performed in Stata 13 software (StataCorp LP, College Station, TX).

## Ethical statement

The research review committee of Chattogram Veterinary and Animal Sciences University, Chittagong, Bangladesh reviewed and approved the study protocol. Written consent was obtained from selected participants before collecting data.

## Results

A total of 415 participants were interviewed; 52.8% were male and the majority of the participants (63.9%) were between the ages of 18 and 30. Most of the participants (79.5%) were came from the Chattogram division. Students were predominant (55.9%) and 52% had bachelor's degrees. More than 35% of the participants' households had a monthly income of more than BDT 30,000 (Table 1).

## Health status of participants

A considerable number of respondents reported that they had a history of familial precedence of cancer (13%), ulcer (17.1%), neurological illnesses (9.9%), hypertension (67.2%) and asthma (27%). Familial infertility problems were reported by 4.3% of respondents, and 6.7% participants' family members experienced vomiting or diarrhea immediately after dried fish consumption within the last 6 months (Table 2).

## Knowledge, attitudes, and practices towards health hazard associated with unsafe dried fish consumption

Overall, 7.2% of the respondents had accurate knowledge of health problems due to unsafe dry fish consumption, 92.8% had a more positive attitude to buying safe dry fish and 26.8% consumed unsafe dry fish more frequently (Table 3). Regarding the present place of residence, there was a significant variation in the knowledge levels among the population ($p = 0.01$). In terms of occupation, there was significant variation in attitude levels among the population ($p = 0.034$). Practice varied significantly within the population of different educational groups ($p = 0.047$) (Table 4).

There was no significant difference in knowledge between males and females (S1 Table). In contrast, there was a significant difference in attitude toward buying dry fish that was stored in an airtight polythene pouch between males and females (S2 Table). Similarly, a significant difference in practice between males and females was found when buying dry fish that are free of flies, insects, or rodents (S3 Table).

**Table 1. Demographic characteristics of the studied population (N = 415, November-December 2022, Bangladesh).**

| Variables | n (%) | 95% CI |
|---|---|---|
| **Age (years)** | | |
| 18–30 | 265 (63.9) | 59.0–68.4 |
| 31–50 | 101 (24.3) | 20.3–28.8 |
| 51–65 | 44 (10.6) | 7.8–14.0 |
| >65 | 5 (1.2) | 0.4–2.8 |
| **Gender** | | |
| Male | 219 (52.8) | 47.8–57.7 |
| Female | 196 (47.2) | 42.3–52.2 |
| **Division** | | |
| Chattogram | 329 (79.3) | 75.1–83.1 |
| Dhaka | 55 (13.3) | 10.1–16.9 |
| Sylhet | 11 (2.7) | 1.3–4.7 |
| Khulna | 7 (1.7) | 0.7–3.4 |
| Rajshahi | 4 (1.0) | 0.3–2.4 |
| Rangpur | 4 (1.0) | 0.3–2.4 |
| Mymensingh | 3 (0.7) | 0.1–2.1 |
| Barisal | 2 (0.5) | 0.1–1.7 |
| **Marital status** | | |
| Married | 160 (38.6) | 33.8–43.4 |
| Unmarried | 248 (59.8) | 54.9–64.5 |
| Widowed | 7 (1.7) | 0.7–3.4 |
| **Education** | | |
| No education | 7 (1.7) | 0.7–3.4 |
| Primary (1–5) | 2 (0.5) | 0.1–1.7 |
| Secondary (6–10) | 36 (8.7) | 6.1–11.8 |
| Intermediate (11–12) | 63 (15.2) | 11.9–19.0 |
| Bachelor | 216 (52.0) | 47.1–57.0 |
| Higher education (above bachelor) | 91 (21.9) | 18.0–26.2 |
| **Occupation** | | |
| Student | 232 (55.9) | 51.0–61.0 |
| Govt. employee | 17 (4.1) | 2.4–6.5 |
| Non-govt. employee | 37 (8.9) | 6.4–12.1 |
| Businessman | 26 (6.3) | 4.1–9.0 |
| Housewife | 67 (16.1) | 12.7–20.0 |
| Unemployed | 10 (2.4) | 1.2–4.4 |
| Others | 26 (6.3) | 4.1–9.0 |
| **Present residence** | | |
| Urban | 339 (81.7) | 77.6–85.3 |
| Semi-urban | 26 (6.3) | 4.1–9.0 |
| Rural | 50 (12.0) | 9.1–15.6 |

## Factors related to knowledge, attitudes and practices

In bivariate analysis, respondents aged between 31 and 50 years were more likely to have more accurate knowledge compared to those aged 18 to 30 years (OR 1.69; 95% CI: 1.04–2.75, *p* 0.034). Females were found to have more accurate knowledge compared to males (OR 1.57; 95% CI: 1.06–2.32, *p* 0.024), and housewives had a higher likelihood of having more accurate

**Table 2. History of family health of the studied population (N = 415, November-December 2022, Bangladesh).**

| Characteristics | N (%) | 95% CI |
|---|---|---|
| **Any of your family members had cancer within the last 10 years?** | | |
| Yes | 54 (13.0) | 9.9–16.6 |
| No | 361 (87.0) | 83.4–90.1 |
| **Any of your family members had ulcer within the last 10 years?** | | |
| Yes | 71 (17.1) | 13.6–21.1 |
| No | 344 (82.9) | 78.9–86.4 |
| **Any of your family members had neurological disorder (autism/ seizure/ dementia/ epilepsy) by born?** | | |
| Yes | 28 (6.7) | 4.5–9.6 |
| No | 387 (93.3) | 90.4–95.5 |
| **Any of your family members currently have neurological disorder?** | | |
| Yes | 41 (9.9) | 7.2–13.2 |
| No | 374 (90.1) | 86.8–92.8 |
| **Any of your family members have hypertension at present?** | | |
| Yes | 279 (67.2) | 62.5–71.7 |
| No | 136 (32.8) | 28.3–37.5 |
| **Any of your family members have asthma at present?** | | |
| Yes | 112 (27.0) | 22.8–31.5 |
| No | 303 (73.0) | 68.5–77.2 |
| **Any of your married family members currently have infertility problem?** | | |
| Yes | 18 (4.3) | 2.6–6.8 |
| No | 397 (95.7) | 93.2–97.4 |
| **Any of your female family members experienced complications during pregnancy or childbirth (i.e., miscarriage)?** | | |
| Yes | 42 (10.1) | 7.4–13.4 |
| No | 373 (89.9) | 86.6–92.6 |
| **Any of your family members experienced vomiting or diarrhea immediately after eating dried fish within the last 6 months?** | | |
| Yes | 28 (6.7) | 4.5–9.6 |
| No | 387 (93.3) | 90.4–95.5 |

**Table 3. Overall knowledge, attitudes, and practices of the studied population (N = 415, November-December 2022, Bangladesh).**

| Factors | n (%) |
|---|---|
| **Knowledge** | |
| More accurate (>9) | 30 (7.2) |
| Moderate (6–9) | 133 (32.1) |
| Less accurate (<6) | 252 (60.7) |
| **Attitudes** | |
| More positive (>20) | 385 (92.8) |
| Moderate (11–20) | 28 (6.8) |
| Less positive (<11) | 2 (0.5) |
| **Practices** | |
| More frequent (>14) | 111 (26.8) |
| Moderate (8–14) | 254 (61.2) |
| Less frequent (<8) | 50 (12.1) |

**Table 4. Test of statistical significances of the variations in respondents' knowledge, attitudes, and practices on health hazards due to unsafe dried fish consumption with their demographic characteristics (N = 415, November-December 2022, Bangladesh).**

| Variables | Knowledge | | | | Attitudes | | | | Practices | | | |
|---|---|---|---|---|---|---|---|---|---|---|---|---|
| | More accurate n (%) | Moderate n (%) | Less accurate n (%) | p-value | More positive n (%) | Moderate n (%) | Less positive n (%) | p-value | More frequent n (%) | Moderate n (%) | Less frequent n (%) | p-value |
| **Age (years)** | | | | | | | | | | | | |
| 18–30 | 20 (4.8) | 91 (21.9) | 154 (37.1) | 0.255 | 245 (59) | 18 (4.3) | 2 (0.5) | 0.920 | 76 (18.3) | 154 (37.1) | 35 (8.4) | 0.362 |
| 31–50 | 5 (1.2) | 25 (6.0) | 71 (17.1) | | 95 (22.9) | 6 (1.4) | 0 (0.0) | | 22 (5.3) | 69 (16.6) | 10 (2.4) | |
| 51–65 | 5 (1.2) | 14 (3.4) | 25 (6.0) | | 40 (9.6) | 4 (1.0) | 0 (0.0) | | 13 (3.1) | 26 (6.3) | 5 (1.2) | |
| >65 | 0 (0.0) | 3 (0.7) | 2 (0.5) | | 5 (1.2) | 0 (0.0) | 0 (0.0) | | 0 (0.0) | 5 (1.2) | 0 (0.0) | |
| **Gender** | | | | | | | | | | | | |
| Male | 19 (4.6) | 78 (18.8) | 122 (29.4) | 0.078 | 204 (49.2) | 13 (3.1) | 2 (0.5) | 0.325 | 59 (14.2) | 128 (30.8) | 32 (7.7) | 0.211 |
| Female | 11 (2.7) | 55 (13.2) | 130 (31.3) | | 181 (43.6) | 15 (3.6) | 0 (0.0) | | 52 (12.5) | 126 (30.4) | 18 (4.3) | |
| **Religion** | | | | | | | | | | | | |
| Islam | 19 (4.6) | 87 (20.9) | 183 (44.1) | 0.497 | 264 (63.6) | 23 (5.5) | 2 (0.5) | 0.762 | 80 (19.3) | 174 (41.9) | 35 (8.4) | 0.813 |
| Hinduism | 11 (2.7) | 44 (10.6) | 63 (15.2) | | 113 (27.2) | 5 (1.2) | 0 (0.0) | | 30 (7.2) | 73 (17.6) | 15 (3.6) | |
| Buddhism | 0 (0.0) | 2 (0.5) | 4 (1.0) | | 6 (1.4) | 0 (0.0) | 0 (0.0) | | 1 (0.2) | 5 (1.2) | 0 (0.0) | |
| Others | 0 (0.0) | 0 (0.0) | 2 (0.5) | | 2 (0.5) | 0 (0.0) | 0 (0.0) | | 0 (0.0) | 2 (0.5) | 0 (0.0) | |
| **Division** | | | | | | | | | | | | |
| Chattogram | 25 (6.0) | 108 (26) | 196 (47.2) | 0.248 | 306 (73.7) | 22 (5.3) | 1 (0.2) | 0.995 | 89 (21.4) | 202 (48.6) | 38 (9.1) | 0.850 |
| Dhaka | 2 (0.5) | 17 (4.1) | 36 (8.7) | | 50 (12.1) | 4 (1.0) | 1 (0.2) | | 14 (3.4) | 33 (8) | 8 (1.9) | |
| Sylhet | 1 (0.2) | 2 (0.5) | 8 (1.9) | | 10 (2.4) | 1 (0.2) | 0 (0.0) | | 1 (0.2) | 8 (1.9) | 2 (0.5) | |
| Khulna | 0 (0.0) | 1 (0.2) | 6 (1.4) | | 6 (1.4) | 1 (0.2) | 0 (0.0) | | 2 (0.5) | 3 (0.7) | 2 (0.5) | |
| Rajshahi | 0 (0.0) | 3 (0.7) | 1 (0.2) | | 4 (1.0) | 0 (0.0) | 0 (0.0) | | 1 (0.2) | 3 (0.7) | 0 (0.0) | |
| Rangpur | 0 (0.0) | 1 (0.2) | 3 (0.7) | | 4 (1.0) | 0 (0.0) | 0 (0.0) | | 1 (0.2) | 3 (0.7) | 0 (0.0) | |
| Mymensingh | 1 (0.2) | 1 (0.2) | 1 (0.2) | | 3 (0.7) | 0 (0.0) | 0 (0.0) | | 2 (0.5) | 1 (0.2) | 0 (0.0) | |
| Barisal | 1 (0.2) | 0 (0.0) | 1 (0.2) | | 2 (0.5) | 0 (0.0) | 0 (0.0) | | 1 (0.2) | 1 (0.2) | 0 (0.0) | |
| **Education** | | | | | | | | | | | | |
| No education | 0 (0.0) | 0 (0.0) | 7 (1.7) | 0.077 | 7 (1.7) | 0 (0.0) | 0 (0.0) | 0.719 | 0 (0.0) | 6 (1.4) | 1 (0.2) | 0.047 |
| Primary (1–5) | 0 (0.0) | 0 (0.0) | 2 (0.5) | | 2 (0.5) | 0 (0.0) | 0 (0.0) | | 0 (0.0) | 2 (0.5) | 0 (0.0) | |
| Secondary (6–10) | 0 (0.0) | 11 (2.7) | 25 (6) | | 33 (8) | 3 (0.7) | 0 (0.0) | | 5 (1.2) | 27 (6.5) | 4 (1.0) | |
| Intermediate (11–12) | 3 (0.7) | 21 (5.0) | 39 (9.4) | | 55 (13.2) | 8 (1.9) | 0 (0.0) | | 14 (3.4) | 39 (9.4) | 10 (2.4) | |
| Bachelor | 17 (4.1) | 80 (19.3) | 119 (28.7) | | 203 (48.9) | 11 (2.7) | 2 (0.5) | | 75 (18.0) | 118 (28.4) | 23 (5.5) | |
| Higher education (above bachelor) | 10 (2.4) | 21 (5.0) | 60 (14.5) | | 85 (20.5) | 6 (1.4) | 0 (0.0) | | 17 (4.1) | 62 (14.9) | 12 (2.9) | |
| **Occupation** | | | | | | | | | | | | |
| Student | 13 (3.1) | 85 (20.5) | 134 (32.3) | 0.143 | 216 (52.0) | 15 (3.6) | 1 (0.2) | 0.034 | 67 (16.1) | 136 (32.8) | 29 (7.0) | 0.131 |
| Govt. employee | 2 (0.5) | 5 (1.2) | 10 (2.4) | | 15 (3.6) | 2 (0.5) | 0 (0.0) | | 7 (1.7) | 8 (1.9) | 2 (0.5) | |
| Non-govt. employee | 5 (1.2) | 8 (1.9) | 24 (5.8) | | 35 (8.4) | 2 (0.5) | 0 (0.0) | | 4 (1.0) | 29 (7) | 4 (1.0) | |
| Businessman | 1 (0.2) | 12 (2.9) | 13 (3.1) | | 23 (5.5) | 3 (0.7) | 0 (0.0) | | 10 (2.4) | 12 (2.9) | 4 (1.0) | |
| Housewife | 4 (1.0) | 14 (3.4) | 49 (11.8) | | 64 (15.4) | 3 (0.7) | 0 (0.0) | | 17 (4.1) | 43 (10.4) | 7 (1.7) | |
| Unemployed | 2 (0.5) | 2 (0.5) | 6 (1.4) | | 8 (1.9) | 1 (0.2) | 1 (0.2) | | 1 (0.2) | 6 (1.4) | 3 (0.7) | |
| Others | 3 (0.7) | 7 (1.7) | 16 (3.9) | | 24 (5.8) | 2 (0.5) | 0 (0.0) | | 5 (1.2) | 20 (4.8) | 1 (0.2) | |
| **Monthly family income (BDT)** | | | | | | | | | | | | |
| <10,000 | 6 (1.4) | 36 (8.7) | 83 (20) | 0.314 | 116 (28) | 9 (2.2) | 0 (0.0) | 0.352 | 27 (6.5) | 83 (20) | 15 (3.6) | 0.534 |
| 10,000–30,000 | 13 (3.1) | 43 (10.4) | 88 (21.2) | | 131 (31.6) | 11 (2.7) | 2 (0.5) | | 40 (9.6) | 88 (21.2) | 16 (3.9) | |
| >30,000 | 11 (2.7) | 54 (13.0) | 81 (19.5) | | 138 (33.3) | 8 (1.9) | 0 (0.0) | | 44 (10.6) | 83 (20) | 19 (4.8) | |
| **Present residence** | | | | | | | | | | | | |

*(Continued)*

**Table 4.** (Continued)

| Variables | Knowledge | | | | Attitudes | | | | Practices | | | |
|---|---|---|---|---|---|---|---|---|---|---|---|---|
| | More accurate n (%) | Moderate n (%) | Less accurate n (%) | *p*-value | More positive n (%) | Moderate n (%) | Less positive n (%) | *p*-value | More frequent n (%) | Moderate n (%) | Less frequent n (%) | *p*-value |
| Urban | 27 (6.5) | 118 (28.4) | 194 (46.8) | 0.011 | 318 (76.6) | 19 (4.6) | 0 (0.0) | 0.256 | 100 (24.1) | 196 (47.2) | 43 (10.4) | 0.003 |
| Semi-urban | 3 (0.7) | 4 (1.0) | 19 (4.6) | | 24 (5.8) | 2 (0.5) | 0 (0.0) | | 8 (1.9) | 15 (3.6) | 3 (0.7) | |
| Rural | 0 (0.0) | 11 (2.7) | 39 (9.4) | | 43 (10.4) | 7 (1.7) | 0 (0.0) | | 3 (0.7) | 43 (10.4) | 4 (1.0) | |
| **Permanent residence** | | | | | | | | | | | | |
| Urban | 17 (4.1) | 56 (13.5) | 95 (22.9) | 0.142 | 156 (37.6) | 10 (2.4) | 2 (0.5) | 0.216 | 55 (13.3) | 87 (21) | 26 (6.7) | 0.027 |
| Semi-urban | 5 (1.2) | 14 (3.4) | 40 (9.6) | | 52 (12.5) | 7 (1.7) | 0 (0.0) | | 15 (3.6) | 38 (9.1) | 6 (1.4) | |
| Rural | 8 (1.9) | 63 (15.2) | 117 (28.2) | | 177 (42.7) | 11 (2.7) | 0 (0.0) | | 41 (9.8) | 129 (31.1) | 18 (4.3) | |

knowledge than students (OR 1.87; 95% CI: 1.03–3.39, *p* 0.039). Participants from rural and urban areas exhibited a significant difference, with the former showing a higher likelihood of having more accurate knowledge (OR 2.75; 95% CI: 1.37–5.53, *p* 0.004). Individuals living in rural areas exhibit a substantially greater tendency (OR 1.89; 95% CI: 1.06–3.36, *p* 0.03) to participate in more frequent practices compared to respondents living in urban areas (Table 5).

Multivariate analysis showed that females had more accurate knowledge (AOR 1.53; 95% CI: 1.03–2.29, *p* 0.0.37) than males. Participants whose present residence was in rural areas were more likely to have more accurate knowledge (AOR 2.64; 95% CI: 1.30–5.36, *p* 0.007) than those whose present residence was in urban or semi-urban areas. Residents of rural areas exhibited significantly more positive attitudes (AOR 4.62; 95% CI: 1.40–15.23, *p* 0.012) compared to their counterparts in urban or semi-urban areas. Respondents with a bachelor's degree were less likely to engage in frequent practices compared to those with no education (AOR 0.19; 95% CI: 0.04–0.87, *p* 0.032). Businessmen were also less likely to engage in more frequent practices compared to students (AOR 0.36; 95% CI: 0.14–0.90, *p* 0.029) (Table 5).

## Discussion

The consumption of unhealthy dried fish can pose significant health risks to individuals and the overall well-being of the community. The study highlights a significant knowledge gap regarding the health risks of unsafe dried fish consumption, despite a positive attitude towards safe alternatives. Rural and urban populations have different knowledge and practices, highlighting the need for tailored public health campaigns. Older participants and females have higher knowledge levels, suggesting they can effectively disseminate information within their communities. The study indicates, a small number of respondents had accurate knowledge of health problems caused by unsafe dry fish consumption but most of the respondents showed positive attitudes toward hazards associated with unsafe dried fish consumption. Many respondents were aware of the potential dangers such as high levels of heavy metals, including mercury, cadmium, and lead in dry fish. Previous research has shown that the majority of heavy metals exhibit cumulative toxicity, accumulating in the human body with repeated exposure [20]. This underscores the importance of being aware of the detrimental effects of these heavy metals. Similar to previous study findings, this study observed a disparity in knowledge levels between rural and urban consumers, where demographic factors influence knowledge acquisition among rural people [21].

Most participants expressed positive attitudes towards purchasing dried fish packaged in airtight polythene pouches. A similar finding was found in another study conducted in Bangladesh, where 82% of consumers indicated a willingness to increase their dried fish consumption

**Table 5. Logistic regression analysis of the variables associated with respondents' knowledge, attitudes, and practices on health hazards due to unsafe dried fish consumption (N = 415, November-December 2022, Bangladesh).**

| Variables | Knowledge | | Attitudes | | Practices | |
|---|---|---|---|---|---|---|
| | OR, 95% CI, *p* | Adjusted OR, 95% CI, *p* | OR, 95% CI, *p* | Adjusted OR, 95% CI, *p* | OR, 95% CI, *p* | Adjusted OR, 95% CI, *p* |
| **Age (years)** | | | | | | |
| 18–30 | Ref. | | Ref. | | Ref. | |
| 31–50 | 1.69, 1.04–2.75, 0.034 | | 0.77, 0.30–1.97, 0.584 | | 1.15, 0.73–1.82, 0.538 | |
| 51–65 | 0.89, 0.47–1.69, 0.730 | | 1.21, 0.39–3.73, 0.737 | | 0.92, 0.49–1.84, 0.796 | |
| >65 | 0.64, 0.13–3.12, 0.578 | | Undefined | | 1.69, 0.33–0.59, 0.526 | |
| **Gender** | | | | | | |
| Male | Ref. | Ref. | Ref. | | Ref. | Ref. |
| Female | 1.57, 1.06–2.32, 0.024 | 1.53, 1.03–2.29, 0.037 | 1.12, 0.53–2.34 0.773 | | 0.87, 0.59–1.27, 0.463 | 0.72, 0.45–1.14, 0.161 |
| **Religion** | | | | | | |
| Islam | Ref. | | Ref. | | Ref. | |
| Hinduism | 0.67, 0.44–1.02, 0.059 | | 0.47, 0.17–1.25, 0.128 | | 1.10, 0.71–1.69, 0.657 | |
| Buddhism | 1.26, 0.24–6.71,0.789 | | Undefined | | 1.03, 0.23–4.72, 0.966 | |
| Others | Undefined | | Undefined | | 1.68, 0.13–21.53, 0.688 | |
| **Division** | | | | | | |
| Chattogram | Ref. | | Ref. | | Ref. | |
| Dhaka | 1.33, 0.74–2.40, 0.338 | | 1.35, 0.49–3.72, 0.561 | | 1.16, 0.65–2.06, 0.617 | |
| Sylhet | 1.70, 0.44–6.52, 0.442 | | 1.32, 0.16–10.78, 0.793 | | 2.25, 0.69–7.36, 0.180 | |
| Khulna | 4.18, 0.50–34.88, 0.187 | | 2.20, 0.25–18.95, 0.474 | | 1.68, 0.33–8.56, 0.530 | |
| Rajshahi | 0.41, 0.07–2.23, 0.299 | | Undefined | | 0.79, 0.12–5.07, 0.802 | |
| Rangpur | 2.14, 0.22–20.34, 0.506 | | Undefined | | 0.79, 0.12–5.07, 0.802 | |
| Mymensingh | 0.23, 0.24–2.24, 0.207 | | Undefined | | 0.18, 0.16–1.95, 0.158 | |
| Barisal | 0.23, 0.01–5.66, 0.371 | | Undefined | | 0.34, 0.02–4.96, 0.431 | |
| **Education** | | | | | | |
| No education | Ref. | | Ref. | | Ref. | Ref. |
| Primary (1–5) | Undefined | | Undefined | | 0.65, 0.03–12.15, 0.774 | 0.80, 0.40–15.85, 0.883 |
| Secondary (6–10) | Undefined | | Undefined | | 0.60, 0.13–2.79, 0.511 | 0.68, 0.14–3.31, 0.631 |
| Intermediate (11–12) | Undefined | | Undefined | | 0.52, 0.12–2.33, 0.394 | 0.49, 0.10–2.31, 0.369 |
| Bachelor | Undefined | | Undefined | | 0.28, 0.07–1.19, 0.084 | 0.19, 0.04–0.87, 0.032 |
| Higher education (above bachelor) | Undefined | | Undefined | | 0.54, 0.13–2.34, 0.412 | 0.44, 0.10–2.01, 0.290 |
| **Occupation** | | | | | | |

*(Continued)*

**Table 5.** (Continued)

| Variables | Knowledge | | Attitudes | | Practices | |
|---|---|---|---|---|---|---|
| | OR, 95% CI, *p* | Adjusted OR, 95% CI, *p* | OR, 95% CI, *p* | Adjusted OR, 95% CI, *p* | OR, 95% CI, *p* | Adjusted OR, 95% CI, *p* |
| Student | Ref. | | Ref. | | Ref. | Ref. |
| Govt. employee | 0.94, 0.35–2.52, 0.900 | | 1.78, 0.38–8.48, 0.467 | | 0.62, 0.23–1.68, 0.348 | 0.33, 0.11–0.97, 0.044 |
| Non-govt. employee | 1.15, 0.56–2.38, 0.699 | | 0.77, 0.17–3.49, 0.733 | | 1.76, 0.89–3.48, 0.107 | 0.98, 0.45–2.11, 0.954 |
| Businessman | 0.79, 0.37–1.70, 0.551 | | 1.75, 0.47–6.44, 0.403 | | 0.76, 0.33–1.74, 0.516 | 0.36, 0.14–0.90, 0.029 |
| Housewife | 1.87, 1.03–3.39, 0.039 | | 0.63, 0.18–2.23, 0.475 | | 1.07, 0.62–1.84, 0.815 | 0.62, 0.31–1.24, 0.178 |
| Unemployed | 0.84, 0.23–3.14, 0.799 | | 3.77, 0.73–19.55, 0.113 | | 3.45, 0.97–12.30, 0.056 | 2.69, 0.71–10.16, 0.144 |
| Others | 1.04, 0.46–2.38, 0.918 | | 1.12, 0.24–5.16, 0.885 | | 1.09, 0.50–2.39, 0.830 | 0.54, 0.23–1.28, 0.162 |
| **Monthly family income (BDT)** | | | | | | |
| <10,000 | Ref. | | Ref. | | Ref. | |
| 10,000–30,000 | 0.76, 0.46–1.25, 0.277 | | 1.30, 0.53–3.14, 0.565 | | 0.79, 0.49–1.27, 0.324 | |
| >30,000 | 0.64, 0.39–1.03, 0.066 | | 0.75, 0.28–2.00, 0.563 | | 0.76, 0.47–1.23, 0.269 | |
| **Present residence** | | | | | | |
| Urban | Ref. | Ref. | Ref. | Ref. | Ref. | |
| Semi-urban | 1.82, 0.74–4.47, 0.190 | 1.90, 0.77–4.71, 0.164 | 1.25, 0.28–5.66, 0.770 | 0.71, 0.13–3.80, 0.688 | 0.92, 0.42–2.056, 0.847 | |
| Rural | 2.75, 1.37–5.53, 0.004 | 2.64, 1.30–5.36, 0.007 | 2.43, 0.98–6.05, 0.056 | 4.62, 1.40–15.23, 0.012 | 1.89, 1.06–3.36, 0.030 | |
| **Permanent residence** | | | | | | |
| Urban | Ref. | | Ref. | Ref. | Ref. | |
| Semi-urban | 1.59, 0.86–2.96, 0.143 | | 1.72, 0.64–4.60, 0.279 | 1.82, 0.61–5.40, 0.282 | 1.11, 0.62–2.01, 0.722 | |
| Rural | 1.35, 0.89–2.05, 0.152 | | 0.80, 0.34–1.86, 0.602 | 0.43, 0.14–1.28, 0.128 | 1.24, 0.81–1.88, 0.317 | |

if packaging and quality were improved [22]. Training is effective in enhancing knowledge of food safety and hygiene [23]. The majority of participants in this study responded positively to participating in consumer awareness training programs about the health risks associated with unsafe dried fish consumption. It was noted that residents of rural areas exhibited significantly more positive attitudes compared to their counterparts in urban or semi-urban areas. This suggests that tailored interventions, such as targeted training and awareness programs, could be particularly effective in rural communities, where positive attitudes towards safe practices are already prevalent.

In this study, many participants reported moderate practices for purchasing dried fish free of flies, insects and rodents and for storing dried fish at home in airtight packets. There has been evidence that improper storage and handling can accelerate lipid rancidity in dried fish, resulting in unpleasant aromas, odors, and hazardous hydroperoxide molecule [24]. Following the purchase of dried fish, consumers should store them in a dry and clean environment. Approximately 77.6% of participants in this study adhere to this practice. Depending on packaging and storage conditions, dried fish can retain its quality for up to 3–6 months [25, 26].

Furthermore, the study revealed interesting patterns related to the frequency of these practices among different demographic groups. Respondents with a bachelor's degree were less likely to engage in frequent practices compared to those with no education. This may suggest a gap in awareness or differing priorities regarding food safety practices among more educated individuals. Similarly, businessmen were less likely to engage in more frequent practices compared to students, which might be attributed to differences in lifestyle, time availability, or access to information about proper dried fish handling and storage.

Globally, one out of ten people get sick from foodborne diseases every year, according to the World Health Organization (WHO), leading to significant morbidity and mortality, with around 420,000 deaths attributed to foodborne diseases [27]. Better knowledge of food safety practices is essential for preventing foodborne illnesses and fostering positive attitudes toward food safety [28, 29]. Despite having awareness of the potential health risks of eating dry fish, some individuals continue to consume dried fish because of its taste, physical appearance, convenience, and trust in sellers [30]. The threats posed by foodborne diseases in developing countries like Bangladesh occur mostly due to improper handling, poor food storage, poor hygiene, inadequate monitoring, poor regulatory systems, and poor awareness [31]. A greater emphasis on public awareness and education campaigns is crucial to raise awareness. By addressing knowledge gaps and promoting safe alternatives, public health efforts can effectively reduce the health risks associated with consuming dried fish. Overall, the findings suggest that there is a need for targeted educational campaigns to improve food safety practices, particularly among younger individuals and students.

## Limitation

This study has several limitations. Social desirability or reporting bias may be present, as individuals may not express their true beliefs or behaviors but instead conform to perceived expectations. The study samples were primarily drawn from populated or significant areas of Chattogram city, excluding several peripheral regions of Chattogram and other districts. The use of purposive sampling may have inadvertently introduced interviewer or selection bias.

## Conclusion

This study highlights the importance of improving knowledge and awareness about the health hazards associated with consuming unsafe dried fish. Most participants in the study had less accurate knowledge about the health hazards of unsafe dried fish. This emphasizes the need for targeted interventions and educational programs to enhance knowledge, attitudes, and practices related to dried fish consumption. Targeted interventions should focus on specific demographic groups, such as women, older age groups, individuals from rural areas, and those with lower education levels. By addressing these factors, it is possible to promote healthier practices and reduce potential health risks associated with the consumption of dried fish.

## Supporting information

**S1 Table. Gender-wise knowledge regarding health hazard due to dried fish consumption (N = 415, November-December 2022, Bangladesh).**
(DOCX)

**S2 Table. Gender-wise attitudes toward health hazard due to dried fish consumption (N = 415, November-December 2022, Bangladesh).**
(DOCX)

**S3 Table. Gender-wise practices of dried fish consumption among the studied population (N = 415, November-December 2022, Bangladesh).**
(DOCX)

**S1 Dataset. Dry fish.**
(XLSX)

## Acknowledgments

The authors express their gratitude to all the participants who generously participated in this study and provided their valuable information.

## Author Contributions

**Conceptualization:** Mahdi Al Hasan Rahat, Anik Saha, Mehedy Hasan Abir, A. S. M. Nafis Sadekeen, Sukanta Chowdury.

**Formal analysis:** Mahdi Al Hasan Rahat, Anik Saha, Mehedy Hasan Abir, A. S. M. Nafis Sadekeen, Sukanta Chowdury.

**Investigation:** Mahdi Al Hasan Rahat, Anik Saha, Mehedy Hasan Abir, Sukanta Chowdury.

**Methodology:** Mahdi Al Hasan Rahat, Anik Saha, Mehedy Hasan Abir, Shahneaz Ali Khan, Sukanta Chowdury.

**Project administration:** Mahdi Al Hasan Rahat, Anik Saha, Mehedy Hasan Abir, A. S. M. Nafis Sadekeen.

**Supervision:** Shahneaz Ali Khan, Sukanta Chowdury.

**Validation:** Shahneaz Ali Khan, Sukanta Chowdury.

**Writing – original draft:** Mahdi Al Hasan Rahat, Anik Saha, Mehedy Hasan Abir, A. S. M. Nafis Sadekeen.

**Writing – review & editing:** Shahneaz Ali Khan, Sukanta Chowdury.

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
