## [Decision Letter · Decision Letter 0]

22 Jul 2024

PONE-D-23-40168Understanding Public Health Risk from Unsafe Dry Fish Consumption in BangladeshPLOS ONE

Dear Dr. Chowdhury,

Thank you for submitting your manuscript to PLOS ONE. After careful consideration, we feel that it has merit but does not fully meet PLOS ONE’s publication criteria as it currently stands. Therefore, we invite you to submit a revised version of the manuscript that addresses the points raised during the review process.

**ACADEMIC EDITOR**:Please kindly address all comments raised by reviewers, not only in your revised manuscript, but also in your responses to their individual comments.

We look forward to receiving your revised manuscript.

Kind regards,

Charles Odilichukwu R. Okpala

Academic Editor

PLOS ONE

Journal Requirements:

2. We notice that your supplementary tables are included in the manuscript file. Please remove them and upload them with the file type 'Supporting Information'. Please ensure that each Supporting Information file has a legend listed in the manuscript after the references list.

**Additional Editor Comments:**

Please, kindly attend to the concerns raised by the reviewers.

Reviewers' comments:

Reviewer's Responses to Questions

**Comments to the Author**

1. Is the manuscript technically sound, and do the data support the conclusions?

Reviewer #1: Yes

Reviewer #2: Partly

2. Has the statistical analysis been performed appropriately and rigorously? 

Reviewer #1: Yes

Reviewer #2: Yes

3. Have the authors made all data underlying the findings in their manuscript fully available?

Reviewer #1: Yes

Reviewer #2: Yes

4. Is the manuscript presented in an intelligible fashion and written in standard English?

Reviewer #1: Yes

Reviewer #2: No

5. Review Comments to the Author

Reviewer #1: 1. Keep keywords in alphabetical order in abstract.

2. In line number66, “There have been no research studies on Bangladesh's food safety problems resulting from the consumption of unsafe dry fish.” How can you say that no research studies yourself?

3. How was the sample size calculated in your study? Explain properly.

Reviewer #2: Introduction

In general the author does not follow the reference style of the journal.

The introduction does not clearly state the extent of the problem of dry fish contamination. It is important to specify the extent to which fish are contaminated with residues and other contaminants that pose public health risks. Relevant studies should be cited to support this information.

Lines 82-84

mention that the study was conducted on people living in Chattogram City, whereas lines 88-89 state, "the study was carried out across multiple divisions in Bangladesh, with a particular focus on the Chattogram division." Please clarify this discrepancy.

Methods

Lines 91-96:

• Were the interviews self-administered or interviewer-administered?

• How were respondents recruited for the web-based questionnaire? What criteria were used, and how was consent obtained?

• Which social media platform was used to administer the questionnaire? This paragraph needs revision for clarity.

Line 105:

• You would wish to use "Health indicators" rather than "Family health history/disease history," as mentioned earlier in line 98.

Lines 116-132:

• How were the cut-offs determined? Is there literature supporting the chosen cut-offs?

• Why did you use Yes/No or Sometimes in judging practices, and how does this translate into More, Moderate, and Less frequent?

Results

Line 149:

• How many participants were interviewed in person versus via the web-based questionnaire? Was there any difference in terms of their responses or data quality?

Lines 154-158:

• Besides vomiting and diarrhea, how are the other diseases related to risks attributed to the consumption of dry fish? If available, please provide relevant information.

Discussion

• Begin with a summary of your findings.

• The discussion section does not mention the results from the multivariate analysis. Critical parts of the results have been omitted from the discussion. Please ensure these results are included and discussed comprehensively.

6. PLOS authors have the option to publish the peer review history of their article (what does this mean?). If published, this will include your full peer review and any attached files.

Reviewer #1: No

Reviewer #2: **Yes: **Agnes Abel Mpinga

---

## [Author Response · Author response to Decision Letter 0]

2 Aug 2024

Thank for sharing the reviewers’ helpful comments and suggestions. Detail responses to each comment are given below for your consideration.

Reviewer: 1

Comment: Keep keywords in alphabetical order in abstract.

Response: Thanks for your comment. Keywords have been rearranged in alphabetical order (line number 37).

Comment: In line number 66, “There have been no research studies on Bangladesh's food safety problems resulting from the consumption of unsafe dry fish.” How can you say that no research studies yourself?

Response: I appreciate your concern. The statement, "There have been no research studies on Bangladesh's food safety problems resulting from the consumption of unsafe dry fish," is based on an extensive literature review conducted during the preparation of this manuscript. While there may be some studies done on dry fish handlers and fishermen, we found no knowledge, attitude, and practice (KAP) studies that have been conducted on consumers in Bangladesh. The sentence has been rewritten to make it clearer (line numbers 66-68). 

Comment: How was the sample size calculated in your study? Explain properly.

Response: Thanks for the comment. The required sample size for this study was 385 individuals based on 95% confidence interval, a 5% margin of error, and the assumption that 50% of the respondents are at risk from consuming unhealthy dried fish. This has been now included in Data collection section (line numbers 100-110).

Reviewer: 2

Comment: In general, the author does not follow the reference style of the journal.

Response: Thank you so much for this comment. The reference style has been changed to “PLoS”. 

Comment: The introduction does not clearly state the extent of the problem of dry fish contamination. It is important to specify the extent to which fish are contaminated with residues and other contaminants that pose public health risks. Relevant studies should be cited to support this information.

Response: Thank you for your insightful comment. I agree that the introduction should clearly state the extent of the problem of dry fish contamination in Bangladesh. To address this, I have added some information from past studies that indicates the harmful effects of dry fish consumption (line numbers 54-56) (line numbers 61-62).

Comment: Lines 82-84: mention that the study was conducted on people living in Chattogram City, whereas lines 88-89 state, "the study was carried out across multiple divisions in Bangladesh, with a particular focus on the Chattogram division." Please clarify this discrepancy.

Response: Thank you for pointing out the inconsistencies. I appreciate your thorough review. Bangladesh has eight divisions, including Chattogram. The study was conducted across these various divisions, with a focus on the coastal division, Chattogram, where the majority of dry fish is produced and consumed. However, dry fish is widely available across the country. To gain full knowledge, the study included these significant divisions. We have revised the content to make this information clear and avoid confusion (line numbers 80-82).

Comment: Lines 91-96: Were the interviews self-administered or interviewer-administered?

Response: Thank you for your comment. The data was collected using two different methods. For in-person interviews, data collection was interviewer-administered. One interviewer initiated the interview and recorded the information in the questionnaire form. In contrast, for online data collection, the process was self-administered. Participants were provided with instructions before filling out the form themselves. We clarified this in the manuscript to ensure the data collection methods are clearly understood (line number 85).

Comment: Lines 91-96: How were respondents recruited for the web-based questionnaire? What criteria were used, and how was consent obtained?

Response: The online questionnaire was distributed through a variety of platforms, including social media, email invitations, and community forums. The selection criteria were being a Bangladesh resident, being over the age of 18, and having completed university. Permission was gathered via a permission form at the start of the online questionnaire, which participants were required to read and agree to before continuing with the survey. Data was mostly collected from bachelor's degree holders who were aware of the importance of filling out forms correctly. Despite caution, social desirability bias or reporting bias can occur, as discussed in the limitations section (line numbers 88-93).

Comment: Lines 91-96: Which social media platform was used to administer the questionnaire? This paragraph needs revision for clarity.

Response: Thank you for your comment. The web-based questionnaire was distributed over several social media channels (Facebook, Instagram, LinkedIn), and Gmail. These platforms were chosen because of their extensive reach and active user base in Bangladesh. We have incorporated this information in the manuscript to help readers understand the recruitment process (line numbers 88-94).

Comment: Line 105: You would wish to use "Health indicators" rather than "Family health history/disease history," as mentioned earlier in line 98.

Response: Thank you for the suggestion. I have revised line 105 to use "health indicators" instead of "family health history/disease history" to ensure consistency with the terminology used in line 98 (line number 97 and 104). 

Comment: Lines 116-132: How were the cut-offs determined? Is there literature supporting the chosen cut-offs?

Response: We are thankful for this observation. We have used some cutoff scores for knowledge, attitudes, and practice questions based on the answers of participants. The correct answer was denoted as 1 and the incorrect answer was denoted as 0, and then we have identified whether each individual’s knowledge is correct or incorrect, attitude is favorable or unfavorable, and practice is good or bad (details are discussed in the “Data collection” section). The cut-offs were set by adopting methods from past KAP studies and also modified to go with our study (please use- https://doi.org/10.21203/rs.3.rs-24562/v2 and doi: 10.12669/pjms.311.6317).

Comment: Lines 116-132: Why did you use Yes/No or Sometimes in judging practices, and how does this translate into More, Moderate, and Less frequent?

Response: Thank you for your comment. The use of "Yes," "No," and "Sometimes" responses was chosen to capture the frequency of specific practices in a simple and straightforward manner. Each response was assigned a numerical value: "Yes" = 2, "Sometimes" = 1, and "No" = 0. This scoring system allowed us to quantify the frequency of practices and calculate a total score for each respondent. The total score ranged from 0 to 22, with higher scores indicating more frequent engagement in the practices.

• Yes: Participants exercise this practice regularly.

• No: The practice is not followed at all.

• Sometimes: The practice is done irregularly.

The cut-off levels were determined as follows:

• More frequent (≥15): Respondents who scored 15 or above frequently engaged in the practices.

• Moderate (8-14): Respondents who scored between 8 and 14 showed moderate engagement.

• Less frequent (≤7): Respondents who scored 7 or below engaged in the practices less frequently.

This method provided a clear and measurable way to categorize the frequency of practices based on respondents' answers

Comment: Line 149: How many participants were interviewed in person versus via the web-based questionnaire? Was there any difference in terms of their responses or data quality?

Response: Thank you for your comment. The study involved 415 participants, including 373 in-person interviews and 42 completing a web-based questionnaire. Initially, we collected data from 435 participants; some of the data was not included in this study as there was incomplete information and they did not satisfy our inclusion criteria. This was done to maintain the integrity of the data.

Concerning the replies and data quality, the following observations were made: 

In-person interviews enabled prompt clarification of questions, resulting in more detailed and consistent responses. Furthermore, the presence of an interviewer ensured that the questionnaire was completed thoroughly, reducing the number of missing data cases.

Web-based questionnaires provided convenience for participants, potentially increasing regional representation. The incomplete information was excluded from the study.

To address potential data quality issues, both datasets underwent rigorous review for completeness and consistency. Any discrepancies were managed according to predefined protocols to uphold the integrity of the data.

Comment: Lines 154-158: Besides vomiting and diarrhea, how are the other diseases related to risks attributed to the consumption of dry fish? If available, please provide relevant information.

Response: Thank you for your comment. Besides vomiting and diarrhea, there are other health effects that can be caused by unsafe dry fish consumption. Aflatoxin B1 (AFB1), T-2 toxin (T-2), ochratoxin A (OTA), and deoxynivalenol (DON) were mycotoxins found to be released by fungi in dried fish products, predominantly by Fusarium, Penicillium, and Aspergillus fungi. If consumed in excess, mycotoxins could cause major health problems like liver cancer, immune issues, and respiratory issues (please use- https://doi.org/10.3390/foods11192938). Moreover, Toxic metal poisoning tends to cause cancer, cardiovascular, brain, kidney, respiratory, reproductive, and neurological problems in human beings, with children been more susceptible to heavy metal toxicity (please use- https://doi.org/10.1016/j.envres.2017.08.051).

Comment: Begin with a summary of your findings. The discussion section does not mention the results from the multivariate analysis. Critical parts of the results have been omitted from the discussion. Please ensure these results are included and discussed comprehensively.

Response: Thank you for your helpful feedback. I revised the discussion section to start with an overview of our findings. Furthermore, I have incorporated the results of the multivariate analysis and ensured that all essential aspects of the results are thoroughly described.

The multivariate analysis findings were highlighted in the updated discussion, providing insights into the major factors impacting participants' views and habits toward dried fish intake. These findings, along with other significant findings, are now thoroughly explored to provide a more complete understanding of the study's consequences (line numbers 192-199) (line numbers 224-226, 234-240 and 252-256).

---

## [Decision Letter · Decision Letter 1]

8 Sep 2024

Understanding Public Health Risk from Unsafe Dry Fish Consumption in Bangladesh

PONE-D-23-40168R1

Dear Dr. Chowdhury,

We’re pleased to inform you that your manuscript has been judged scientifically suitable for publication and will be formally accepted for publication once it meets all outstanding technical requirements.

Kind regards,

Charles Odilichukwu R. Okpala, PhD

Academic Editor

PLOS ONE

Additional Editor Comments (optional):

A very great work done, acceptable for publication.

Reviewers' comments:

Reviewer's Responses to Questions

**Comments to the Author**

1. If the authors have adequately addressed your comments raised in a previous round of review and you feel that this manuscript is now acceptable for publication, you may indicate that here to bypass the “Comments to the Author” section, enter your conflict of interest statement in the “Confidential to Editor” section, and submit your "Accept" recommendation.

Reviewer #1: All comments have been addressed

Reviewer #2: All comments have been addressed

2. Is the manuscript technically sound, and do the data support the conclusions?

Reviewer #1: Yes

Reviewer #2: Yes

3. Has the statistical analysis been performed appropriately and rigorously? 

Reviewer #1: Yes

Reviewer #2: Yes

4. Have the authors made all data underlying the findings in their manuscript fully available?

Reviewer #1: Yes

Reviewer #2: Yes

5. Is the manuscript presented in an intelligible fashion and written in standard English?

Reviewer #1: Yes

Reviewer #2: Yes

6. Review Comments to the Author

Reviewer #1: Authors have addressed all comments as asked by the reviewers. I recommend this manuscript for the publication.

Reviewer #2: The authors should in the limitations include the challenges observed with the use of the web-based questionnaire

In line 154-158: Give brief clarification of the diseases that are indirectly caused by fish consumption such as immune issues, cardiovascular etc..

7. PLOS authors have the option to publish the peer review history of their article (what does this mean?). If published, this will include your full peer review and any attached files.

Reviewer #1: No

Reviewer #2: **Yes: **AGNES ABEL MPINGA

---

## [Editor Report · Acceptance letter]

13 Sep 2024

PONE-D-23-40168R1 

PLOS ONE

Dear Dr. Chowdury, 

I'm pleased to inform you that your manuscript has been deemed suitable for publication in PLOS ONE. Congratulations! Your manuscript is now being handed over to our production team.

Kind regards, 

on behalf of

Dr. Charles Odilichukwu R. Okpala 

Academic Editor

PLOS ONE